

# Visual resource extraction and artistic communication model design based on improved CycleGAN algorithm

Anyu Yang[1] and Muhammad Kashif Hanif[2]

[1] International School of Arts, Dalian University of Foreign Languages, Dalian, Liaoning, China
[2] Department of Computer Science, Government College University, Faisalabad, Pakistan

## ABSTRACT

Through the application of computer vision and deep learning methodologies, real-time style transfer of images becomes achievable. This process involves the fusion of diverse artistic elements into a single image, resulting in the creation of innovative pieces of art. This article centers its focus on image style transfer within the realm of art education and introduces an ATT-CycleGAN model enriched with an attention mechanism to enhance the quality and precision of style conversion. The framework enhances the generators within CycleGAN. At first, images undergo encoder downsampling before entering the intermediate transformation model. In this intermediate transformation model, feature maps are acquired through four encoding residual blocks, which are subsequently input into an attention module. Channel attention is incorporated through multi-weight optimization achieved *via* global max-pooling and global average-pooling techniques. During the model's training process, transfer learning techniques are employed to improve model parameter initialization, enhancing training efficiency. Experimental results demonstrate the superior performance of the proposed model in image style transfer across various categories. In comparison to the traditional CycleGAN model, it exhibits a notable increase in structural similarity index measure (SSIM) and peak signal-to-noise ratio (PSNR) metrics. Specifically, on the Places365 and selfi2anime datasets, compared with the traditional CycleGAN model, SSIM is increased by 3.19% and 1.31% respectively, and PSNR is increased by 10.16% and 5.02% respectively. These findings provide valuable algorithmic support and crucial references for future research in the fields of art education, image segmentation, and style transfer.

# INTRODUCTION

In tandem with the advancement of society and the relentless march of scientific and technological progress, there exists an elevated yearning among the populace for artistic representations, aligning with the gradual refinement of non-material culture's quality of life. An ardent desire prevails for the ability to craft artworks echoing the stylistic resonance of renowned masterpieces—a feat that remains elusive for those devoid of

Corresponding author
Anyu Yang, y13941124025@163.com

painting prowess (*Wang, Li & Vasconcelos, 2021*). The extraction of visual resources is crucial for the dissemination of art. By extracting visual elements such as color, shape, composition, *etc.* from artwork, it helps to convey the artist's intentions and emotions, enhance the audience's perception and understanding of the work, deepen communication and emotional resonance between the artwork and the audience, and further promote the dissemination and influence of art. Image style migration stands as a prominent branch within the domain of computational vision, with its fundamental concept centered upon the transmutation of one image's aesthetic style into that of another while simultaneously preserving the intrinsic structural essence of the original image to the utmost degree. By harnessing the formidable computational prowess of computers, this endeavor proliferates the capacity for style migration across a multitude of images, even facilitating the emulation of diverse painting styles through the artistry of AI technology (*Kwon & Ye, 2022*).

The neural network model, a creation that mimics the interconnections among neurons in the human brain and the principles governing information transmission, has emerged as a quintessential construct in the landscape of technology. The deep neural networks, born from the tenets of deep learning theory, exhibit intricate and profound neuron linkages, more faithfully mirroring the cognitive dynamics of the human brain. In contrast to rudimentary neural networks, their deep counterparts possess a heightened capacity for learning, and trained deep neural networks wield a formidable aptitude for generalization. This profound duality of learning and generalization endows deep neural networks with widespread utility, permeating domains such as text analysis, language processing, image manipulation, and numerous other realms (*Singh et al., 2021*).

Currently, there has been a gradual emergence of style migration methods rooted in the realm of deep neural networks, capturing the attention of numerous scholars who have embarked on the exploration of deep learning techniques to facilitate the transmutation of artistic painting features into photographic representations, thereby yielding entirely novel photographic compositions. Deep neural networks serve as the instrumental conduits for the extraction and translation of these artistic style attributes, endowing the designated images with these features, thus birthing fresh imagery brimming with artistic allure. Deep convolutional neural networks (CNNs) have demonstrated unparalleled proficiency in addressing a multitude of computer vision tasks, transcending the abilities of contemporary humans and encompassing domains such as image classification and target detection (*Singh et al., 2021*). Their prowess extends not only to the realm of static images but also transcends into more intricate and demanding scenarios. As the field of image processing delves deeper into the abyss of exploration, visionary scholars have unveiled novel concepts and perspectives in the domain of image style migration. In recent years, an array of network architectures, including VGGnet, AlexNet, ResNet, and Neural Style Transfer Network, have been conceived and established to enhance the sphere of image processing. In the domain of image style transference, VGGNet typically serves as the prime feature extractor, deftly capturing both content and stylistic nuances through feature extraction at various levels within the image. Simultaneously, AlexNet proves instrumental in analogous feature extraction tasks, focusing primarily on capturing content-related information within the image (*Wang et al., 2015*). Meanwhile, residual networks (ResNet), with its

capacity for more profound feature extraction, excels in enhancing the conveyance of content nuances within the image, yielding superior results. Moreover, the introduction of generative adversarial networks (GANs) has heralded a new era in image generation research. Leveraging the artistry of image generation techniques to craft new images imbued with distinctive artistic styles based on existing visual compositions currently stands as a vibrant focal point within the realm of research (*Li et al., 2020*).

In light of the prevailing disconnect between various artistic styles and their integration within the realm of art education and correlated research, this article introduces an image style migration framework, underpinned by GAN networks, to bridge the gap in artistic style congruence within the art model propagation process, while also addressing the challenge of style conversion. The contributions of this article are delineated as follows:

1. Pioneering an image style migration technique rooted in deep learning GAN networks, this approach transforms image styles, thereby enriching available resources for art education and other relevant applications.

2. Introducing an image-style migration model that employs an attention mechanism in conjunction with the CycleGAN network, this model undergoes pre-training and comprehensive testing.

3. By utilizing the proposed image style migration framework to execute style transmutations across diverse datasets, including public data and self-created datasets, the empirical results unequivocally confirm the superiority of the methodology outlined in this article. This is demonstrated by surpassing several performance metrics compared to existing methodologies. In the remainder of this article, related work is described in 'Related works'. 'Methodology' established the ATT-CycleGAN model. Experiment results and related analysis are detailed described in 'Experiment', and 'Discussion' is the Discussion. The conclusion is drawn at last.

## RELATED WORKS

As visual resource extraction is so important to artistic communication, to design the new communication model, the original information should be enhanced. In this article, we will review the related works about image style migration research and the methods used in this area.

### Image style migration

Image style migration is an innovative technique designed to transmute the style of a given image into that of another designated content image. This approach offers the flexibility to apply diverse artistic styles to images, enabling the application of landscape-style paintings to ordinary photographs, thereby realizing a transformative filtering function. The origins of image-style migration techniques can be traced back to the 1990s when the focus gradually shifted towards their development. Most texture migration algorithms employ non-parametric methods to synthesize image textures while simultaneously preserving the image's underlying structure, utilizing a variety of distinct techniques. *Efros & Freeman (2001)* pioneered the incorporation of texture synthesis concepts to generate stylized images. *Frigo et al. (2016)* introduced an instance-based partitioning approach that varied

patch sizes and sought to identify the optimal patch matches between the target image and the reference image. *Elad & Milanfar (2017)* proposed a classical patch-matching-based style migration technique with adaptive patch enhancements aimed at augmenting image quality. Traditional style migration algorithms adeptly simulate specific artistic styles but face limitations when extending to other, unexplored styles. For novel styles, a considerable investment of time is required to analyze the nuances of the style, drawing upon substantial human knowledge and experience. Furthermore, traditional style migration approaches predominantly extract the foundational features of an image. Consequently, the primary drawback of conventional style migration techniques lies in their inability to produce satisfactory stylized images when dealing with images characterized by intricate color palettes and complex textures Before the advent of deep learning, the methodologies employed for image style migration were primarily centered around image rendering. These image rendering techniques are often categorized into stroke rendering, region rendering, and instance rendering, relying on image processing filter techniques (*Hertzmann et al., 2001*). Although these rendering methods effectively bestow style upon content images, they predominantly operate on the image's underlying information, often failing to encapsulate the abstract features that genuinely encapsulate the content's essence.

The ascent of CNN models within the domain of image processing has been characterized by remarkable milestones, commencing with AlexNet in 2012 and culminating with VGGNet in 2014 and ResNet in 2015. As the CNN model matures post-2016, style migration techniques increasingly adopt deep learning to facilitate the transmutation of image styles. *Gatys, Ecker & Bethge (2015)*, in their groundbreaking study, discerned the feasibility of segregating image content and style representations across different layers of CNN. Building upon this insight, they advanced an image iteration-based style migration algorithm (*Gatys, Ecker & Bethge, 2016*). *Johnson, Alahi & Fei-Fei (2016)* on the other hand, introduced a swift image style migration technique, wherein the desired stylized image is generated *via* the construction of an image generation network. These endeavors collectively underscore the evolving landscape of style migration, marked by the growing integration of diverse neural network methodologies aligned with the burgeoning capabilities of deep learning technology. Consequently, the expansion of research applications in the realm of style migration through deep learning networks has surged to the forefront of contemporary research endeavors, affirming its status as a compelling and vibrant area of exploration in recent years.

## Image migration method based on generative adversarial networks

GANs ascertain the fidelity of generated samples by learning the underlying data distribution, thus addressing the challenge of not having exact sample matches. Instead of generating an average outcome from all reasonable samples, GANs strive to generate a more plausible sample within the generative space. In 2014, *Goodfellow et al. (2014)* introduced the GAN model, comprising a generator (G) and a discriminator (D). GANs optimize both the generator and discriminator through adversarial training, whereby the generator endeavors to produce the most convincing results. *Radford, Metz & Chintala (2017)* proposed DCGANs, which integrate GANs with deep CNN architectures and are

tailored for specialized image generation. _Chen, Lai & Liu (2018)_ presented CartoonGAN, designed to produce cartoon-style images by incorporating edge facilitation loss. This technique can generate diverse cartoon images when provided with a real scene as the content image. _Isola et al. (2017)_ introduced CGAN-based image-to-image translation, requiring paired data during the training process. _Zhu et al. (2017)_ developed CycleGAN, a model that eliminates the need for paired images and enables image-to-image conversion. _Hu, Ding & Li (2020)_ devised a style migration model based on GAN, abandoning the redundant structure of two GAN models trained by CycleGAN. Instead, they employ feature maps from the VGG network to constrain the semantic content between the input image and the generated image. This approach streamlines training, with only one generator and one discriminator being employed, thereby saving time. Style migration plays a pivotal role in various fields, including art, film and television post-production, and game scene development. CycleGAN, DiscoGAN, and DualGAN, all rooted in the concept of dyadic learning, are parallel efforts in the realm of GAN (_Yi et al., 2017_). These models enable style migration tasks with just the dataset, transcending the need for image pairing and significantly expanding the application scope of style migration. _Choi et al. (2018)_ proposed the StarGAN model, which can achieve the transfer of multiple styles within a single GAN. Although the StraGAN model can achieve multiple styles of transfer, the number of styles is still limited, and cannot achieve the transfer of any style. _Xu et al. (2020)_ combined the AdaIN module with the generative adversarial network and added the AdaIN module to the adversarial network, enabling the network to generate images of any style. _Cho et al. (2019)_ combined WCT with GAN and proposed the GDWCT model, which approximates WCT by using regularization and grouping calculations while reducing the number of parameters.

The studies outlined above vividly illustrate the substantial progress made in the domain of deep learning-based generative content research, particularly concerning images. Deep learning-based style migration techniques have exhibited impressive advancements, significantly enhancing migration speed and migration quality. While various GAN networks have seen improvements, their fundamental frameworks have remained relatively consistent (_Kim et al., 2017_). Consequently, in this study, CycleGAN is selected as the foundational network model, upon which enhancements and reinforcement training are employed specifically for character images and landscape images. This approach is undertaken with the goal of bolstering the applicability of these techniques to forthcoming research endeavors in the domains of art education and dissemination.

## METHODOLOGY

After reviewing related works on image style migration and the application of CycleGAN, we will introduce the employed method, including CNN and ResNet, CycleGAN, and the established model.

### CNN and ResNet

CNN and ResNet stand as pivotal neural network architectures within the domain of deep learning, primarily harnessed for computer vision tasks, including image classification,

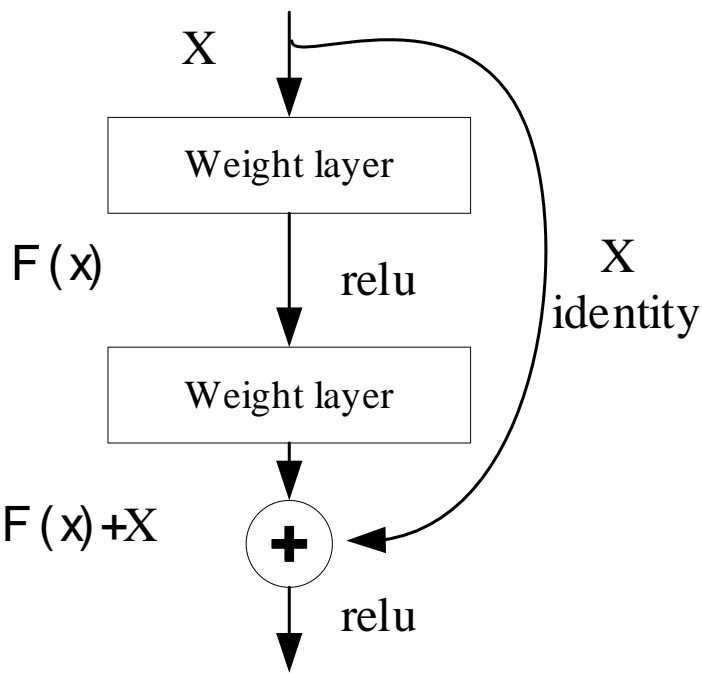

**Figure 1 The structure for the ResNet.**

target detection, and image segmentation. CNNs, specifically, are purpose-built deep learning models adept at processing data with a grid-like structure, a prime example being images. They autonomously acquire salient image features through convolutional and pooling layers, subsequently leveraging fully connected layers to undertake tasks such as classification or regression. The core operation in the convolutional phase is encapsulated in Eq. (1):

$$C(x,y) = \sum i \sum j I(x+i, y+j)K(i,j) \tag{1}$$

where C is the output after the convolution, and I(x,y) is the position of the image and K is the position of the kernel. Traditional CNN networks often suffer from gradient explosion and network is too deep and difficult to converge. For this reason, ResNet was proposed based on residual features (*Wightman, Touvron & Jégou, 2021*). ResNet's fundamental concept revolves around facilitating the training of deep neural networks by enabling the direct transfer of information between different network layers through skip connections or jump connections. This structural design simplifies the network's ability to effectively capture and learn a wide array of features, rendering it highly successful in image classification and a multitude of other computer vision tasks (*Iqbal & Ali, 2018*). The residual block, serving as the foundational building block, is visually depicted in Fig. 1:

The block consists of two weight layers and a jump join. Assuming the input is x, the output of the residual block can be expressed as:

$$F(x) = H(x) + X \tag{2}$$

where F(x) is the output, x is the input, and H(x) is the residual function, which represents the residuals to be learned by the network. By adding x to H(x), the residual block enables the network to learn the residuals. In terms of gradient, by converting from cumulative multiplication to cumulative addition, the effectively overcomes the problem of gradient dispersion, which can be analyzed by Eq. (3):

$$X_L = X_l + \sum_{i=1}^{L-1} F(x_i, w_i) \tag{3}$$

where $X_L$ represents the gradient propagation for the next step and F is the output of the last output after the convolution.

## Attention enhanced CycleGAN for the image style migration generation

The GAN embodies the concept of a zero-sum game, as rooted in game theory. In this GAN framework, two principal entities engage in a strategic game: the Generator (G) and the Discriminator (D) (*Din et al., 2020*). The primary objective of the Generator is to create synthetic data that closely matches the distribution of real sample data. Subsequently, the Generator submits both the generated data and genuine data to the Discriminator, which must discern whether the presented data originates from real or generated sources. In generative adversarial networks (GANs), generators create realistic synthetic data from random noise, aiming to deceive discriminators, which, in turn, strive to accurately distinguish between real and generated data. The adversarial interplay between these components leads to continuous improvement, culminating in the generation of high-quality, realistic data. In this process, the discriminator struggles to differentiate between real and synthetic samples, demonstrating the overall efficacy of GANs in data synthesis. The adversarial interplay between the Generator and the Discriminator in the GAN framework is formally captured in Eq. (4):

$$\min_G \max_D V(D, G) = E_{x \sim P_{data}(x)}[\log D(x)] + E_{z \sim P_z(x)}[\log(1 - D(G(x)))] \tag{4}$$

where x represents the real data, and z is noise, and G(z) is the generated data, and $P_{data}(x)$ is the real data distribution, and $P_z(x)$ denotes the noise distribution.

For the discriminator D, the cost function $J^{(D)}$ is of the form shown below:

$$J^{(D)}\left(\theta^{(D)}, \theta^{(G)}\right) = -\frac{1}{2} E_{x \sim P_{data}} \log D(x) - \frac{1}{2} E_{x \sim P_z} \log(1 - D(G(z))) \tag{5}$$

A zero-sum game is played between the generator and the discriminator, the combined cost of the two is zero, and the cost functions of the generator and the discriminator are shown in Eq. (6):

$$J^{(G)} = -J^{(D)}. \tag{6}$$

Thus it can be represented by a value function V to represent $J^{(G)}$ and $J^{(D)}$.

$$V\left(\theta^{(D)}, \theta^{(G)}\right) = E_{x \sim P_{data}} \log D(x) + E_{x \sim P_z} \log(1 - D(G(z))) \tag{7}$$

$$J^{(G)} = -J^{(D)} = \frac{1}{2}V\left(\theta^{(G)}, \theta^{(D)}\right) \tag{8}$$

By finding a suitable V-value function to make the cost function as small as possible, from completing the definition of the value function as a problem of solving very large and very small values. That is, the most discriminants D⋆ and G for the problem shown in Eq. (8) are found:

$$\underset{G}{\arg\min}\ \underset{D}{\max}\ V(D, G) \tag{9}$$

CycleGAN, akin to GAN, operates as an unsupervised learning model, facilitating the migration of image styles across distinct domains. The model's innovation lies in its ability to learn mappings between different domains without the need for paired training data. The integration of an attention mechanism further refines its performance by focusing on crucial regions during style translation, preserving intricate details. The inclusion of pre-training and comprehensive testing ensures robust generalization to diverse datasets and artistic styles. Through these features, CycleGAN accomplishes effective image style migration, making it a valuable tool for applications such as art education, where the unsupervised learning approach enables the model to autonomously acquire and replicate diverse artistic styles. The CycleGAN model, along with the DiscoGAN and DualGAN models, stands out for not requiring dataset matching to achieve cross-domain mapping relationships and the transmutation of different styles between domains. Within the CycleGAN architecture, two mapping functions are at play: G and F, complemented by two distinct discriminators, DX and DY. Here, X and Y denote two image datasets originating from different domains. The Generator G serves as the mapping from domain X to domain Y, while Generator F serves as the counterpart, mapping from domain Y to domain X (*Gnanha et al., 2022*).

CycleGAN uses a cyclic consistency loss such as in Eq. (10) $L_{cyc}$ to represent this difference.

$$
\begin{aligned}
L_{cyc}\left(G_{c \to p}, G_{p \to c}\right) = &\ E_{c p_{data}(c)}\left[\left\|G_{p \to c}\left(G_{c \to p}(c)\right) - c\right\|_1\right] \\
&+ E_{p p_{data}(p)}\left[\left\|G_{c \to p}\left(G_{p \to c}(p)\right) - p\right\|_1\right]
\end{aligned} \tag{10}
$$

where, the $G_{c \to p}$ denotes the forward generator *i.e.,* G, and $G_{p \to c}$ denotes the backward generator *i.e.,* F. The cyclic process of CycleGAN is shown in Fig. 2:

The generator structure under this network structure is shown in Fig. 3:

The generative network initiates its operation by downsampling the image through an encoder before entering the intermediate transformation model. Within the intermediate transformation model, the image undergoes four encoding residual blocks to derive the feature map. Subsequently, the feature map is channeled into the attention module, where the attention mechanism is introduced at the encoding stage. The integration process of the attention module into the encoding stage is visually depicted in Fig. 4:

As illustrated in Fig. 4, this attention-weighting module introduces attention through channel weighting. Initially, the encoded feature map undergoes global average pooling (GAP) and global maximum pooling (GMP) procedures, enabling the extraction of both global and fine-grained texture information from the image, respectively. Global average

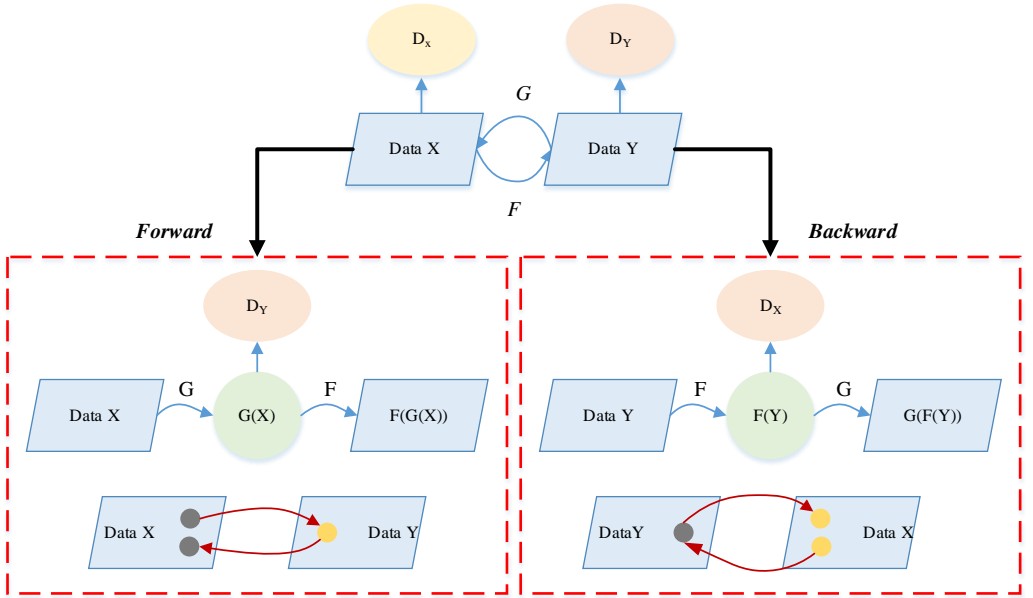

**Figure 2** **The cycle consistency for the CycleGAN consistency.**

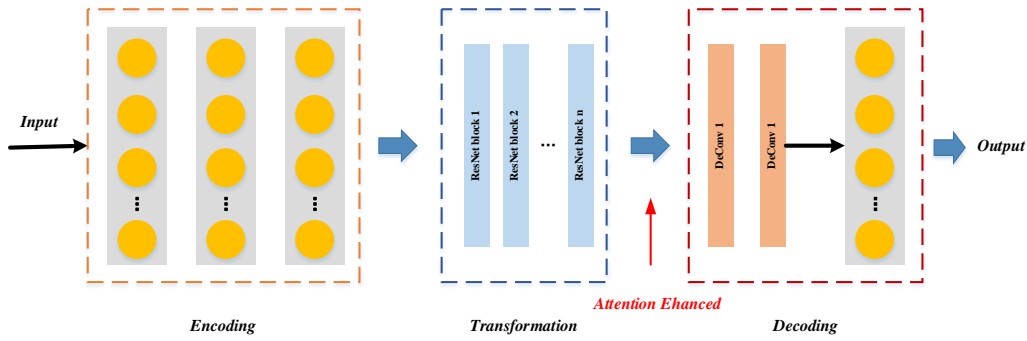

**Figure 3** **The generator G structure.**

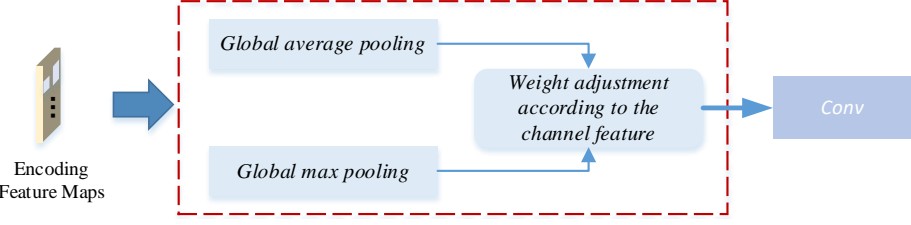

**Figure 4** **The attention mechanism for the CycleGAN.**

**Figure 5** The framework for the image style migration in art evaluation.

pooling is achieved through sliding window averaging across the feature map, while global maximum pooling involves selecting the maximum value for each feature map Finally, the feature map is multiplied by learned parameter weights, assigning distinct weights to each channel within the encoder feature map. The magnitude of these weights determines the channel's significance in the feature, effectively introducing the attention mechanism (*Yan et al., 2022*).

Building upon this foundation, we establish an evaluation framework for the image style migration model based on the proposed ATT-CycleGAN. The overall framework is outlined in Fig. 5.

Style migration through ATT-CycleGAN is realized using pertinent public datasets and the art teaching images employed in this study.

## EXPERIMENT

### Datasets

For the experimental data, two distinct publicly available datasets were chosen to facilitate comparative testing in this study. Given the specific focus on style migration within the domain of art, the primary emphasis is placed on style transformation tasks and the migration of landscape object styles. To that end, the selfie2anime dataset (*Torbunov et al., 2023*) is utilized for data analysis in the style transformation process, while the image styles are acquired from the Places365 dataset (*Zhou et al., 2017*) for both training and testing purposes. The Places365 dataset is a comprehensive scene recognition dataset encompassing over 1.8 million images across 365 scene categories (https://zenodo.org/records/5926442). In this article, we select a subset of 1,000 paintings from the dataset, as provided by Svoboda, for model testing. The selfie2anime dataset, on the other hand, comprises selfies and comics (https://zenodo.org/records/10130356). For the selfie component, over 40,000 selfies are available, with only women's photos utilized for training. The anime section involves over 20,000 face images, and for this dataset, female anime faces from the manga section are chosen. Similarly, for this data selection, 1,000 samples are selected for training and testing. Additionally, 100 images are specifically curated, segregating them into sketch style and original images to align with the design requirements of the art teaching and communication model. The information for these datasets can be summarized as shown in Table 1. All dataset splits are executed at a 7:3 ratio.

To assess the model's performance, classic bilinear interpolation (*Gribbon & Bailey, 2004*), the approach proposed by *Johnson, Alahi & Fei-Fei (2016)*, the method proposed by

**Table 1 The summarized information for the employed datasets.**

| Dataset | Data size | Style |
| --- | --- | --- |
| selfie2anime | More than 40K | Self-portrait and cartoon |
| Places365 | More than 1800k | Landscape and Ink Painting |

*Han et al. (2018)*, as well as the original unprocessed CycleGAN, are selected for method comparisons in model validation.

## Experiment setup and details

Structural similarity index measure (SSIM) and peak signal-to-noise ratio (PSNR) metrics are used to evaluate the model performance during the selection process of evaluation metrics (*Hore & Ziou, 2010*), and the above performances are calculated as shown in Eqs. (11) and (12):

$$\text{SSIM}(X, Y) = \frac{(2\mu_X\mu_Y + C_1)(2\sigma_{XY} + C_2)}{(\mu_X^2 + \mu_Y^2 + C_1)(\sigma_X^2 + \sigma_Y^2 + C_2)} \tag{11}$$

where X and Y denote the images to be measured, the $\mu_X$ and $\mu_Y$ denote the images X and image Y are the mean values of the $\sigma_X^2$ and $\sigma_Y^2$ represent the mean values of the images X and the covariance of Y the covariance of $C_1$ and $C_2$ The value of SSIM ranges from 0 to 1. The closer the value is to 1, the higher the similarity between the two images, and the more serious the image distortion is. PSNR is an important index for evaluating the quality of the image, and its calculation is shown in Eq. (12).

$$\text{MSE} = \frac{1}{H \times W} \sum_{i=1}^{H} \sum_{j=1}^{W} (X(i,j) - Y(i,j))^2$$
$$\text{PSNR} = 10 \times \log 10 \left( \frac{(2^n - 1)^2}{\text{MSE}} \right) \tag{12}$$

where, the MSE denotes the image X and image Y The mean square error between H and W denote the height and width of the image; and n denotes the number of bits per pixel, and n The larger the value of PSNR, the smaller the distortion of the image, which means that the image has higher quality.

After completing the selection of data sets for the model and the establishment of relevant evaluation indicators, the modeling algorithm of our proposed method is as follows:

## Experiment results and analysis

After completing the training process, as outlined in Algorithm 1, and obtaining the respective models based on the selected dataset and the self-constructed dataset, we conducted model testing and analysis. In the context of model comparison, a comprehensive evaluation was carried out, encompassing the unimproved CycleGAN model, along with the methodologies discussed in Section 'Datasets'. The results obtained from the selfie2anime dataset are shown in Table 2 and Fig. 6.

Based on the SSIM and PSNR metrics employed in the article, it is evident that the model proposed in this study achieves a more balanced and improved performance

---

**Algorithm 1: Training process of ATT-CycleGAN for the Image style migration**

---

**Input:** The collected image, selected Places365 and selfie2anime

**Initialization.**

Define the **ATT- CycleGAN**.

Weights initialization.

**Initialization**: set batch size, learning rate, epochs, weights, bias.

**Model training:** Epochs initialization.

**while** epoch<preset epoch **do**

Sample data from Input.

Feed data to the proposed model.

Model updates.

**End**

**Parameters Fine tuning**

**while** epoch<*preset epoch* **do**

Validation dataset input.

Loss calculation.

**Compute** SSIM and PSNR

**Save** the optimal model

**end**

**Output:** Trained **ATT-CycleGAN** network

---

**Table 2  The result comparison among different methods on selfie2anime.**

| Method | Bilinear interpolation | Johnson et al. | Chu et al. | CycleGAN | Proposed |
|---|---|---|---|---|---|
| SSIM | 0.791 | 0.769 | 0.839 | 0.841 | 0.852 |
| PSNR | 18.75 | 16.38 | 18.64 | 19.13 | 20.09 |

**Notes.**
*Johnson, Alahi & Fei-Fei, 2016*.
*Chu & Shih, 2013*.

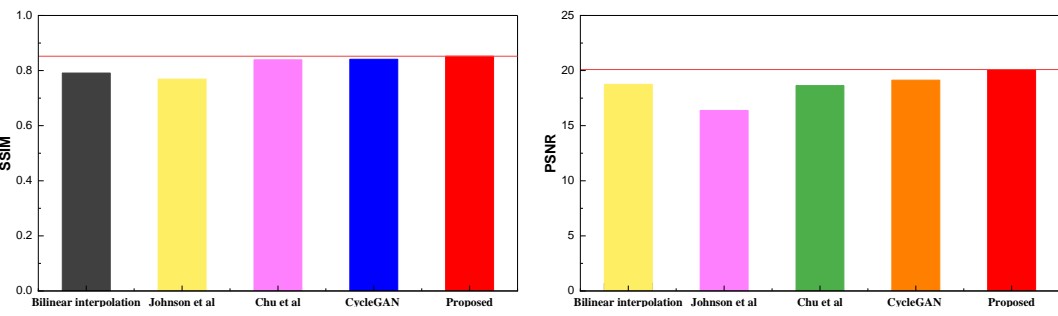

**Figure 6  The result comparison among different methods on selfie2anime.**

after the integration of the attention mechanism. The results across the two metrics demonstrate a superior and more consistent performance. After conducting the analysis

**Table 3  The result comparison among different methods on Places365.**

| Method | Bilinear interpolation | Johnson et al. | Chu et al. | CycleGAN | Proposed |
|---|---|---|---|---|---|
| SSIM | 0.697 | 0.583 | 0.857 | 0.846 | 0.873 |
| PSNR | 15.79 | 17.85 | 19.38 | 20.86 | 22.98 |

Notes.
_Johnson, Alahi & Fei-Fei, 2016_.
_Chu & Shih, 2013_.

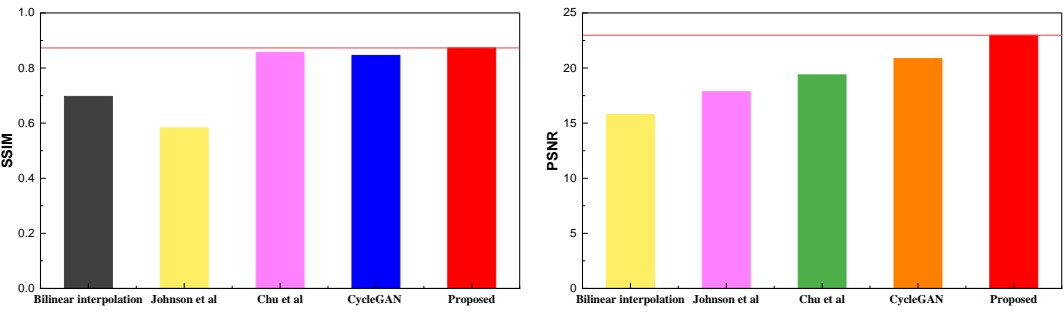

**Figure 7  The result comparison among different methods on Places365.**

**Table 4  The result comparison among in different methods on self-established dataset.**

| Method | Bilinear interpolation | Johnson et al. | Chu et al. | CycleGAN | Proposed |
|---|---|---|---|---|---|
| SSIM | 0.701 | 0.687 | 0.853 | 0.849 | 0.872 |
| PSNR | 16.39 | 17.03 | 17.34 | 17.58 | 19.37 |

Notes.
_Johnson, Alahi & Fei-Fei, 2016_.
_Chu & Shih, 2013_.

on the selfie2anime dataset, a similar analysis was extended to the Places365 dataset, with the obtained results presented in Table 3 and Fig. 7.

The comparative results presented above highlight the superior performance of the method proposed, which extends to the Places365 dataset as well. The higher SSIM scores under the Places365 dataset compared to the selfie2anime dataset suggest that the proposed method excels in handling landscape-class images. This underscores its robust image analysis capabilities, particularly when applied to more complex visual content. After concluding the analysis of the two public datasets, an additional analysis was conducted on the self-constructed dataset, with the results depicted in Table 4 and Fig. 8.

The self-constructed datasets featured exclusively consist of landscape images. The outcomes from these datasets closely align with those of the Places365 dataset, signifying the inherent advantages of CycleGAN-like models for handling data with higher pixel counts and richer visual content. Furthermore, the proposed method maintains its superiority over other methods even when an attention mechanism is integrated. Additionally, this

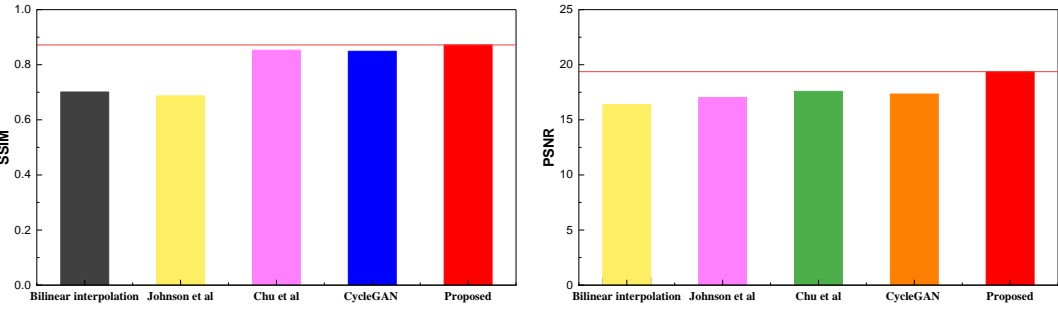

**Figure 8 The result comparison among in different methods on self-established dataset.** *Johnson, Alahi & Fei-Fei, 2016* and *Chu & Shih, 2013*.

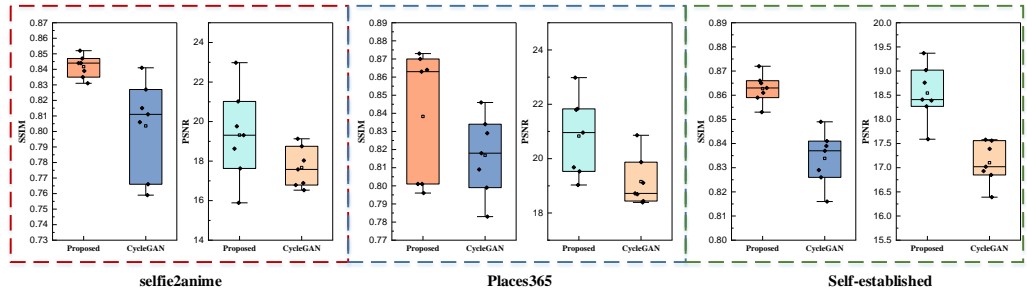

**Figure 9 The performance test for the proposed model and CycleGAN with different batch sizes.**

study conducted data testing with varying batch sizes, and the results are visually presented in Fig. 9,

Figure 9 primarily serves to contrast the performance of the proposed method with the CycleGAN method, which lacks the attention mechanism across different datasets and varying inputs. The boxplot visualizations illustrate that, following the integration of the attention mechanism, the proposed method consistently outperforms the counterpart model in terms of both SSIM and PSNR metrics. These findings affirm that the proposed method excels in both performance and stability, showcasing the advantages of the attention mechanism's incorporation.

## DISCUSSION

This article introduces the ATT-CycleGAN network, which incorporates an additive attention mechanism, for the exploration of image style migration and automatic image generation within the realm of art education and dissemination. The foundational network architecture of CycleGAN is employed as the base for pixel image style migration. Addressing the challenge of maintaining linear details in the generated pixel images during the migration process, the generator structure of CycleGAN is enhanced. Weight addition is automated by calculating the channel count, ensuring the method robustness. In the model comparison process, various techniques are assessed, including CycleGAN and basic

methods like BI, without the attention mechanism. Particular attention is given to methods such as *Johnson, Alahi & Fei-Fei (2016)* and Chu (*Torbunov et al., 2023*), focusing on their performance in the intended application conditions outlined in this article. Experimental results underscore that, with the inclusion of the attention mechanism, CycleGAN excels in the migration of photography style and portrait style, offering a novel avenue for art teaching and dissemination. Furthermore, this approach extends the range of applicable data. In conclusion, the incorporation of an attention mechanism into the CycleGAN network presents a powerful tool for investigating image style migration in the context of art communication education. This approach is poised to have a profound impact on the fields of art and education, fostering development and innovation in both domains.

The remarkable performance of deep learning techniques in the realm of computer vision has substantially reduced the reliance on human resources and effectively tackled numerous challenges that were prevalent in traditional methods. As generative network technology continues to mature, areas such as image style migration have seen widespread utilization, offering extensive opportunities for future research. Notably, CycleGAN networks, enhanced by the inclusion of the attention mechanism, demonstrate the potential to achieve more precise style migration while preserving image content. This enhancement serves as a technical foundation for prospective applications in the movie media industry, game development, VR, AR, and various other domains. Art style migration, in particular, holds the promise of enabling a comprehensive understanding of an artist's unique artistic traits. Simultaneously, it accelerates the efficiency of art media dissemination, thereby fostering greater creativity and expression.

## CONCLUSION

This article introduces an intelligent image style migration and image generation model based on ATT-CycleGAN, aiming to provide algorithmic support for style research and migration challenges within the domain of art education. The article details the incorporation of multi-channel attention through Global Average Pooling (GAP) and Global Maximum Pooling (GMP) in the CycleGAN generator, enabling the extraction of richer global and texture information from images. After enhancing the network, training and testing for image style migration were conducted using the selfie2anime and Places365 datasets, focusing on portraits and landscapes. The results demonstrate the network's superior performance, as reflected in SSIM and PSNR metrics. Furthermore, actual data tests were conducted for the style migration conversion of landscape images, yielding favorable results in both quantitative metrics and qualitative manual evaluations. Among them, the SSIM index of 0.872 is better than the single CycleGAN index of 0.849, while under the PSNR index, the result of 19.37 is still better than the single CycleGAN index of 17.58. The methodological framework presented herein offers valuable insights and technical support for future research endeavors in art education and image style conversion.

In future research, efforts will be directed towards enhancing the model's generalization performance by addressing more complex challenges, such as the horse2zebra dataset,

alongside avatar and landscape data. This includes the pursuit of multi-task objectives, encompassing target segmentation recognition and style migration. Additionally, expanding the dataset diversity to accommodate a broader range of image transformation styles represents another key research objective.

### Funding
The authors received no funding for this work.

### Competing Interests
The authors declare there are no competing interests.

### Author Contributions
- Anyu Yang conceived and designed the experiments, performed the experiments, performed the computation work, prepared figures and/or tables, authored or reviewed drafts of the article, and approved the final draft.
- Muhammad Kashif Hanif analyzed the data, performed the computation work, prepared figures and/or tables, and approved the final draft.

### Data Availability
The code are available in the Supplemental Files.

The Polish places dataset is available at Zenodo:

Konstantinos Stavridis, Athanasios Psaltis, Anastasios Dimou, & Petros Daras. (2022). Polish Places dataset (1.0) [Data set]. Zenodo. https://doi.org/10.5281/zenodo.5926442.

The selfie2anime dataset is available at Zenodo:

Kim. (2023). selfie2anime dataset [Data set]. Zenodo. https://doi.org/10.5281/zenodo.10130356.

### Supplemental Information
Supplemental information for this article can be found online at http://dx.doi.org/10.7717/peerj-cs.1889#supplemental-information.

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
