# Peer review of "Visual resource extraction and artistic communication model design based on improved CycleGAN algorithm"

_PeerJ Computer Science, doi:10.7717/peerj-cs.1889_

## Round 0.1 · original submission · Major Revisions

Both reviews provide constructive feedback on the reporting, experimental design, and validity of the findings in a paper focused on intelligent image style transfer and image generation based on ATT CycleGAN. The reviewers highlight issues such as inconsistencies in the introduction section, insufficient explanations for formulas, and the need for additional evaluation indicators in the experimental part. Moreover, it suggests incorporating recent journal articles for better references. The reviewers acknowledge favorable results but recommend refining abstract keywords, adding introductory paragraphs to sections for clarity, clarifying the role of visual resource extraction, explaining the functions of generators and discriminators in GAN frameworks, and reorganizing content for a more logical flow. It also emphasizes the importance of providing concrete numerical representations in conclusion and improving overall language expression and modification throughout the paper. Combining these insights, the paper could benefit from addressing these issues to enhance its coherence, clarity, and credibility.

**Language Note:** The review process has identified that the English language must be improved. PeerJ can provide language editing services - please contact us at [email protected] for pricing (be sure to provide your manuscript number and title). Alternatively, you should make your own arrangements to improve the language quality and provide details in your response letter. – PeerJ Staff

·

Basic reporting

1. In the related works, when introducing the background, it is necessary to strengthen the relationship between thematic visual resources and art communication models.
2. The author's main contribution format in the introduction section is inconsistent, and I would suggest that the author change it to the same sentence format.
3. The content in section 2.2 is too small to explain the relevant model methods and the relevant content can be logically rearranged after adding.
4. In the third section, some formulas lack corresponding explanations, and I would suggest the author add relevant introductions.
5. What are the specific advantages of CycleGAN compared to Gans used in the paper? What does it accomplish with unsupervised learning?

Experimental design

6. The experimental part seems to lack some other evaluation indicators, such as accuracy, which is not conducive to illustrating the generalization ability of the model.
7. The presentation of the data set and experimental parameters in section 4.1 is somewhat confusing and can be presented in tabular form.

Validity of the findings

8. This paper lacks some excellent articles from recent journals as citations, so it is necessary to select some excellent articles for reference

Reviewer 2 ·

Basic reporting

This paper introduces an intelligent image style transfer and image generation model based on ATT CycleGANFavorable results were obtained in both quantitative and qualitative manual evaluations. But I would suggest that the author make the following changes to improve the paper:
1. The abstract keywords require refinement to better align with the main content of the article. The revised keywords should accurately represent the core themes and contributions of the research.
2. To enhance reader comprehension, it is recommended that the author incorporates an introductory paragraph at the beginning of each section. This introductory paragraph should succinctly outline the primary content and objectives of the respective section, providing readers with a clear roadmap.

Experimental design

3. What is the direct help and function of visual resource extraction for an art communication model? Clarify the direct utility and functional role of visual resource extraction within the context of an art communication model. Elaborate on how visual resource extraction contributes to the overall effectiveness of the art communication model.
4. How do generators and discriminators work in GAN frameworks, what roles do they play, and where do they focus their efforts? Provide a comprehensive explanation of the functions and operations of generators and discriminators within the Generative Adversarial Network (GAN) frameworks. Clearly define their roles, interactions, and specific areas of focus in contributing to the overall efficacy of GANs.
5. The procedure of modeling Algorithm 1 in Part 4 should not be placed in the experimental part, I would recommend placing it in Part 3. This adjustment ensures a more logical flow and organization of the content, aligning the modeling process with the theoretical foundation.
6. The content of the experiment is too little. Augment the experiment section by incorporating ablation experiments. This addition will strengthen the model interpretation, providing a more robust understanding of the model's performance under various conditions.

Validity of the findings

7. In the conclusion section, the numerical representation should be given according to the concrete results of the experiment. This addition ensures that the conclusion is supported by specific and measurable outcomes, enhancing the credibility of the research findings.
8. Address instances where language expression and modification may be insufficient throughout the paper. Strengthen the overall quality of writing by refining and polishing the language to meet professional standards. Pay particular attention to clarity, coherence, and precision in conveying ideas and results.

Annotated reviews are not available for download in order to protect the identity of reviewers who chose to remain anonymous.

---

## Round 0.2 · accepted · Accept

Both reviewers have confirmed that the authors have addressed all of their comments.

·

Basic reporting

no comment

Experimental design

no comment

Validity of the findings

no comment

Reviewer 2 ·

Basic reporting

It looks good, no comment.

Experimental design

No comment

Validity of the findings

No comment